# EmoSign: A Multimodal Dataset for Understanding Emotions in American Sign Language

## Abstract

Unlike spoken languages where the use of prosodic features to convey emotion is well studied, indicators of emotion in sign language remain poorly understood, creating communication barriers in critical settings. Sign languages present unique challenges as facial expressions and hand movements simultaneously serve both grammatical and emotional functions. To address this gap, we introduce EmoSign, the first sign video dataset containing sentiment and emotion labels for 200 American Sign Language (ASL) videos. We also collect open-ended descriptions of emotion cues, such as specific expressions and signing speed, that lead to the identified emotions. Annotations were done by 3 Deaf ASL signers with professional interpretation experience. Alongside the annotations, we include benchmarks of baseline models for sentiment and emotion classification. Our benchmark results show that current multimodal models fail to integrate visual cues into emotional reasoning and exhibit bias towards positive emotions. This dataset not only addresses a critical gap in existing sign language research but also establishes a new benchmark for understanding model capabilities in multimodal emotion recognition for sign languages. This work can inspire new architectures that integrate fine-grained visual understanding with linguistic context awareness to distinguish e.g., syntactic versus affective functions of visual cues.

## 1 Introduction

The emotional content of speech comes not only from its linguistic content but also *how* it is spoken– from pitch and intonations to non-verbal expressions (Cutler et al., 1997). For sign language, however, emotional indicators are less understood and studied (Elliott & Jacobs, 2013; Hietanen et al., 2004; Lim et al., 2024). This ambiguity has practical negative consequences from misinterpretations of signers' feelings to causing biases and prejudices in legal settings (Taira & Itagaki, 2019) and emergency departments (James et al., 2022). Recent advances in multimodal emotion recognition (Zadeh et al., 2018; Chen et al., 2017; Poria et al., 2018) and emotion-aware language models (Lian et al., 2025; Li et al., 2024a; Cheng et al., 2024) have shown promising results for spoken languages, where incorporating emotional understanding improves performance on affective content recognition and translation tasks Chen et al. (2024); Yang et al. (2025); Song et al. (2023). However, these approaches have not been extended to sign languages.

Emotion recognition in sign languages presents unique technical challenges that distinguish it from conventional multimodal emotion recognition. Unlike spoken languages where emotional and linguistic information can be separated across different channels (prosody vs. words), sign languages embed both functions within the same visual modalities. For example, facial expressions simultaneously serve grammatical markers (e.g., raised eyebrows for yes/no questions) and emotional indicators (Valli & Lucas, 2000; Lim et al., 2024). Shorter sentences, more angular sign movements, and shortened movement paths are observed when individuals express anger (Reilly et al., 1992). This creates a fundamental modeling challenge: systems must disentangle these dual functions rather than treating all facial expressions as emotional cues. The scarcity of emotion-labeled sign language data compounds these difficulties. Existing American Sign Language (ASL) datasets 1 focus primarily on translation tasks with English captions, lacking the emotional annotations necessary for developing and evaluating affect recognition systems.

To address this critical research gap, we introduce EmoSign, the first comprehensive dataset containing sentiment and emotion labels for ASL videos. The dataset includes 200 ASL video clips

annotated by 3 Deaf ASL signers with professional interpretation experience, who provided: (1) overall sentiment ratings on a 7-point scale, (2) presence and intensity ratings for 10 distinct emotion categories (3) detailed descriptions of specific emotion cues. Unlike existing sign language datasets that focus primarily on translation capabilities, EmoSign specifically targets the affective dimensions of signing. Our benchmark results with 4 multimodal models show that current models fail to integrate visual cues and heavily rely on text captions for emotion reasoning. This lack of visual grounding is especially important in the context of sign languages as the emotional nuances come from visual cues. These models also exhibit bias towards positive emotions.

Our work contributes to both sign language accessibility and emotion recognition research by: (1) providing the first dedicated dataset for studying emotional expression in ASL; (2) documenting descriptions of how emotions manifest through manual and non-manual components through the lens of native signers; and (3) establishing baseline model performance for sentiment analysis and emotion classification tasks in sign language. Beyond sign language emotion recognition, our findings have broader implications for multimodal models. The poor performance observed in vision-only condition suggests a need for specialized visual encoders trained specifically on non-verbal emotional expression, rather than relying on general-purpose vision models. Additionally, our ablation experiments motivate the development of architectures that prevent text from overwhelming visual information. Finally, our dataset also provides a foundation for future work on emotion recognition without linguistic shortcuts, potentially including non-verbal domains like mime and dance. Code and data will be made publicly available after acceptance.

## 2  RELATED WORK

**Multimodal approach to emotion recognition.** Recent advances in large language models (LLMs) have expanded the scope of emotion recognition beyond the traditional paradigm of emotion label prediction. These models facilitate a more generative approach to emotion *understanding*, producing detailed, comprehensive descriptions of emotional states in natural language (Lian et al., 2025). This shift has prompted the development of new datasets and metrics that accommodate rich natural language descriptions of emotions, allowing for greater nuance in emotion analysis. Increasingly, multimodal approaches that combine video, audio, text and image inputs are being explored as a way of improving model robustness in complex environments (Lee et al., 2019; Lian et al., 2023). However, many existing multimodal models are still heavily reliant on (Liang et al., 2024b), or biased towards (Xiao et al., 2024), the language modality. To address this limitation, recent work has begun investigating novel sources of data that capture nonverbal social cues without relying on language, such as mime videos (Li et al., 2025). Sign language videos are also a rich source of expressive nonverbal data. They present unique challenges, but also opportunities, for multimodal emotion recognition. In hearing communities, many emotional expressions are universal in the sense that they can be reliably understood by people across many different cultures (Cordaro, 2014). However, in sign languages, facial expressions and other non-manual components (such as mouth shapes and body language) often simultaneously serve grammatical and emotional functions. As such, recognizing the emotion of sign videos is often challenging for non-signers (Lim et al., 2024).

**Machine learning research on sign language.** Prior machine learning work on sign language has focused on sign language translation and production. Sign language translation (SLT) methods typically use either raw image data or skeletal representation of the signer's pose as input (Boháček & Hrúz, 2022) or they rely on gloss-annotated datasets (gloss: a method of sign language labeling). The translation capabilities of LLMs appear to extend to sign languages (Fang et al., 2024). Recent work has developed gloss-free sign language translation methods using large multimodal models, with LLaVA-SLT (Liang et al., 2024a) demonstrating that hierarchical visual encoders and linguistic pretraining can approach gloss-based translation accuracy without requiring expensive gloss annotations. In parallel with SLT, research has also investigated sign language production (SLP), which typically involves converting text to gloss, mapping the gloss to pose, then rendering the pose into a video or avatar (Fang et al., 2024). Despite significant advances in both SLT and SLP, challenges remain in developing systems that accurately capture and convey emotional nuance in sign language. Existing sign language datasets mainly capture English captions and sometimes gloss (Table 1). Closest to our work is FePh(Alaghband et al., 2020), where facial expressions of sign language videos were annotated. While FePh also aims to capture emotions expressed by signing, it differs from our EmoSign dataset in several important ways: (1) The raw data used for FePh was cropped

to the signer's face. However, facial expressions are not the only parameter used by signers to express emotions—the size and speed of signing as well as body language also play an important role (Reilly et al., 1992). (2) FePh appears to have hired hearing annotators to label the images. This could potentially be problematic, as hearing individuals frequently misinterpret signers' facial expressions (Lim et al., 2024). (3) FePh contains only binary labels (presence/absence) of emotions. In comparison, EmoSign contains labels for more fine-graned annotations for sentiment and emotional intensity. In addition, to the best of our knowledge, our dataset is the first sign language dataset to include qualitative descriptions of how emotions manifest in signing, from the perspective of native signers.

# 3 EmoSign: Multimodal Dataset of Sign Language with Emotion Labels

Collecting a dataset of this nature is challenging due to several important factors: (1) Sign language data cannot be simply obtained through platforms like Prolific, and requires building trust with the D/deaf and Hard-of-Hearing community. The authors spent months interviewing community members, attending events, learning ASL, and collaborating with experts at Deaf universities and departments. (2) Our dataset required unique expertise, specifically Deaf native signers with professional interpretation experience who can distinguish grammatical from emotional facial expressions. Such cultural and linguistic nuance in ASL requires native fluency. (3) Each utterance required three layers of annotation (sentiment, emotion categories, and open-ended cue descriptions), which is substantially more complex than typical emotion labeling tasks. Finally, our dataset is unique in this area. To the best of our knowledge, we are the first to establish benchmarks and baseline metrics for emotion recognition in ASL, enabling evaluations that are not available in other existing benchmarks. Considering the cost of time and budget, we start with 200 utterances. High-quality datasets that are similar in size have proven valuable for benchmarking (Arodi et al., 2024; Krojer et al., 2024; Li et al., 2024b). Figure 1 overviews the dataset construction process. Next, we describe how we prepared the dataset for annotation, built the annotation pipeline, and post-processed the annotations.

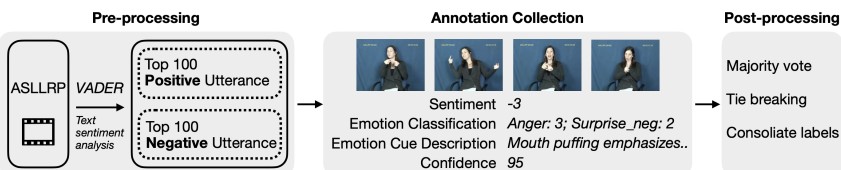

Figure 1: Overview of dataset construction process. We used VADER to filter for emotionally salient videos based on text captions. Each utterance was then annotated by 3 deaf native ASL signers, and final annotation for each clip was selected based on majority vote.

## 3.1 Dataset Collection and Pre-processing

We considered recording ASL videos where signers would be instructed to act out contain specific emotional content, but we chose existing ASL clips instead for scientific and practical reasons. Acted emotions in affective computing research may not represent how emotions are expressed in everyday communication McKeown et al. (2011), and using a well-documented corpus enables reproducibility and comparison with future studies.

Table 1 summarizes existing continuous signing ASL datasets. We excluded YouTube-ASL due to the uncertainty of the quality of the signing and captions in the dataset. For the remaining datasets, we sampled several hours of video from each and segmented the videos into sentence or utterance-long clips. We then used VADER (Hutto & Gilbert, 2014), a lexicon and rule-based sentiment analysis tool, to predict the text sentiment of each clip's caption. Across the board, we found that a large majority of videos were associated with neutral or close-to-neutral text captions as most videos were from news broadcasts and educational contents. Based on this preliminary review, we selected ASLLRP (Neidle et al., 2022) as the base dataset as it provides the most comprehensive and high-quality labels for each video and also contains videos with strong emotional intensity.

Table 1: Overview of existing ASL datasets. EmoSign is the only dataset that contains fine-grained emotion and sentiment labels and emotion cue descriptions annotated by Deaf native signers.

| Dataset | Size | Signers | ASL Fluency | Source | Labels |
|---|---|---|---|---|---|
| YouTube-ASL (Uthus et al., 2023) | 984h | >2500 | Mixed/unknown | Web | English captions |
| OpenASL (Shi et al., 2022) | 288h | 220 | DHH and interpreters | Web | English captions |
| ASL STEM Wiki (Yin et al., 2024) | >300h | 37 | Interpreters | Lab | English sentences |
| ASLLRP (Neidle et al., 2022) | 2,651 utterances | 19 | ASL native signers | | English and gloss captions, non-manual information |
| How2Sign (Duarte et al., 2021) | 79h | 11 | Interpreters | Lab | English captions |
| MS-ASL (Joze & Koller, 2018) | 24h | 222 | Mixed/unknown | Web | English caption |
| **EmoSign** | 16 min, 200 utterances | 3 | Deaf native signers | Lab | English and gloss captions, Emotion & sentiment labels |

We began by manually inspecting ASLLRP's continuous signing videos for emotionally expressive text content, then segmented these videos into utterances (typically consisting of a single sentence or phrase) based on the utterance start and end frames provided in the dataset. As before, we used VADER to calculate the text sentiment of each utterance. Despite the pre-selection for emotionally expressive videos, a large proportion of utterances were still associated with neutral or close-to-neutral text captions. To achieve emotional representation in the dataset, we selected the 100 most positive and 100 most negative utterances based on the VADER scores for the final dataset.

## 3.2 ANNOTATION CONSTRUCTION

This study was reviewed and approved by our Institutional Review Board. We recruited 3 ASL native signers through a third-party vendor to annotate the selected video clips (for more details about recruitment, see Appendix A.1) . Prior to beginning the annotations, the signers attended a training session with the researchers to walk through the annotation process, align on the expected responses for each annotation task, and clarify and questions they may have regarding the tasks. The annotation took roughly 3 hours per individual. The annotations were collected via Qualtrics (See Appendix 4 for details about the annotation interface). The interface was refined through pilot tests with individuals whose first language is ASL (excluded from the final team of annotators).

For each video, the signers completed three annotation tasks in the following order: (1) sentiment analysis, (2) emotion classification, and (3) free response description of attributions of emotions. For the general sentiment, the annotators were asked to determine the overall sentiment of the video from strongly negative to strongly positive on a linear scale from $-3$ to $+3$ (Zadeh et al., 2018). For emotion classification, the annotators were asked to determine the level of presence of 10 emotions from "not present at all" to "extremely present" on a scale of 0 to 3. The set of emotions are "joy", "excited", "surprise (positive)", "surprise (negative)", "worry", "sadness", "fear", "disgust", "frustration", and "anger". The labels built on Ekman's basic emotions, (Ekman, 1992) and the circumplex model of affect (Russell, 1980), and was also informed by prior work on emotion datasets that expand on basic emotion categories for richer annotations (Liu et al., 2022).

After the first two tasks, the annotators were asked to rate their confidence in their scores on a scale of 0-100 (0: not confident at all; 100: extremely confident). Finally, the annotators were asked to describe specific cues that led them to identify the emotions they chose. Example descriptions were given about the speed and scale of movement, head and body movement, facial expressions, and signs that were emphasized. These guided questions were derived based on prior literature and interview results (Chua et al., 2025) as well as our pilot tests. Note that we allow the annotators to skip any videos that they did not wish to annotate because of the content or the quality of the videos.

## 3.3 DATASET POST-PROCESSING

Table 2: Krippendorff's Alpha scores of each label of the Dataset on a scale of -1 to 1, where 1 indicates unanimous agreement and -1 indicating systematic disagreement.

| label | alpha | label | alpha | label | alpha |
|---|---|---|---|---|---|
| Sentiment | 0.738 | surprise_neg | 0.119 | disgust | 0.166 |
| joy | 0.699 | worry | 0.555 | frustration | 0.330 |
| excited | 0.552 | sadness | 0.333 | anger | 0.370 |
| surprise_pos | 0.381 | fear | 0.351 | **average** | 0.593 |

Each clip was labeled by minimally 1, maximally 3 annotators, given a very small fraction of the clips were skipped. For each label, we used the "majority vote" approach to find the most popular

rating. In the case of a tie, we selected the label from the most confident annotator. Table 2 shows the Krippendorff's alpha scores Krippendorff (2011) used to measure the agreement between the annotators of each label. Overall, the average score is 0.593. Within the emotion categories, positive emotion labels had higher inter-annotator agreement than negative emotion labels. To contextualize, existing widely-used emotion recognition datasets had lower inter-annotator agreement compared to ours: MELD (Fleiss' kappa = 0.43) (Poria et al., 2018), IEMOCAP (Fleiss' kappa = 0.48) (Busso et al., 2008).

## 3.4 FINAL DATASET ANALYSIS

The final dataset includes 200 utterances with an average length of 4.8s per utterance and a total of about 16 minutes of video. Figure 2 A shows the distribution of the duration of clips. The dataset includes 4 different signers, and primarily depicts scenarios from everyday life such as conversations about the weather, family members and medical checkups.

Figure 2 B shows the distribution of the sentiment labels. There are relatively few clips with neutral sentiment, but this is expected, since we selected clips with captions that had salient positive or negative emotions based on VADER. Figure 2 C shows the distributions of the emotion categories. We binarized the presence of each emotion and the detailed breakdown distribution can be found in Appendix A.5.

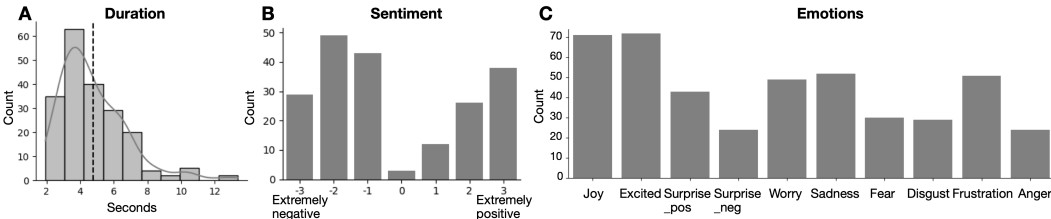

Figure 2: (A) Duration distribution of the clips in the dataset. Dashed line indicates mean. (B) Distribution of sentiment labels. Labels correspond to a 7-point Likert scale where -3: Extremely negative, 0: Neutral, and 3: Extremely positive. (C) Distribution of emotion categories based on binarized presence across clips.

Looking at the annotators' response to attributions of emotional cues, we found some common themes which we elucidate in brief: (1) The non-manual markers are the primary cue for recognizing emotion in ASL. These includes *facial expressions* such as furrowed brows, pursed lips, and squinted eyes, *head movements* such as the head thrusts, tilting, and orientation changes that intensify the emotion, *mouth movements* such as "O" shape, tongue out, and puffed lips, and *body movement* such as shoulder raising and full-body tilting. (2) Signs were modified and emphasized for emotional expressions. Sign size (large/small), speed (fast/slow), repetition, and finger-spelling (sometimes for emphasis) are all noted as sign-based emotional markers. For stronger emotions (both positive and negative), signs are produced more broadly, quickly, or emphatically, very often in parallel with expressive non-manuals markers. (3) The role and context of the sentence is important to disambiguate emotions. Markers such as shifts in eye gaze, physical orientation, and changes in signing space are all used by signers to signal narrative switching or changes in perspective. However, the lack of broader context can cause uncertainty in emotion identification.

## 4 BENCHMARKS

### 4.1 BENCHMARK TASKS

We describe three tasks of increasing complexity that we carried out using the EmoSign dataset. Sentiment analysis and emotion classification are common emotion recognition tasks, while the emotion cue grounding task allows us to understand the mechanisms of multimodal models through their reasoning behind each prediction.

**Sentiment Analysis**. In this classification task, the goal is to predict the overall sentiment of a sign video on a 7-point Likert scale (-3 to 3), with a score of -3 corresponding to Strongly Negative

and a score of +3 corresponding to Strongly Positive. In addition, we also formulated a coarse version of the task where the model is simply required to predict whether the sentiment is positive, neutral or negative. Following prior work (Zadeh et al., 2018; 2017), we evaluate performance using accuracy (ACC) and weighted F1-score (wF1). We chose wF1 as the primary metric and ACC as the secondary metric due to label imbalance.

**Emotion Classification**. Building on the previous task, the emotion classification task aims to predict the emotions present in a sign video. We first combined 'joy' and 'excited' into one category, 'happiness' due to their high co-occurrence within a single video (Jaccard similarity score of 0.81, Figure 5 in Appendix). If all of the emotion categories were labeled as 'not present', then a 'neutral' label is assigned. We separated the dataset into two subsets: a single-expression set containing videos with a single dominant emotion (140 clips), and a multi-expression set containing videos with muliple emotions present (37 clips, filtered to exclude combinations only present in a single sample.) Appendix A.5 shows the distributions of each subset.

For the single-label classification task, the model is required to select a single emotion from a pre-defined set of ten possible labels (described in Section 3.2), plus the label Neutral. The model is not required to assess emotional intensity in this task. For each single-expression label, we present the accuracy and F1 score. Holistically, we chose weighted accuracy (wAcc) and weighted F1-score (wF1) as the evaluation metric based on prior approaches (Jiang et al., 2020; Liu et al., 2022).

**Emotion Cue Grounding**. The ability to accurately identify task-relevant temporal and spatial regions of a video, otherwise known as grounding, is a crucial component of video question-answering (Min et al., 2024) systems and has a wide range of valuable applications (Wadley et al., 2022; Yang et al., 2012). Yet, there remains a significant performance gap between MLLMs and human annotators on visual grounding tasks (Xiao et al., 2024). The goal of the emotion cue grounding task is to identify video frames and spatial regions relevant to the sentiment analysis and emotion classification tasks described above.

## 4.2 EXPERIMENTAL SETUP

We selected four multimodal LLMs (MLLMs) that support video-language inputs to obtain baseline results on the EmoSign dataset: GPT-4o, AffectGPT (Lian et al., 2025), Qwen2.5-VL-7B-Instruct (Bai et al., 2025) and MiniGPT4 (Ataallah et al., 2024). Model cards are provided in Appendix A.2. For all models and tasks, we conduct ablation studies through 3 conditions (caption-only, video-only, and video+caption as input) to better understand the influence of individual modalities. Inference was conducted on a single 80GB NVIDIA Tesla A100 GPU.

Each video was sampled at 10 fps and coded in base64. We chose 10 fps as there is no significant intelligibility loss for ASL isolated signs from 30 to 10 fps, and most of the energy of SL motion may lie below 6 or 7 Hz (Bigand et al., 2021). For GPT-4o, the model temperature was set to zero, and we forced structured output where each API calls outputs responses to all three tasks (See Appendix A.3 for prompts and sample outputs). In preliminary tests, we found that AffectGPT, Qwen2.5 and MiniGPT4 were unable to consistently produce clean output when prompted to respond to all three benchmark tasks at once. We adapted the prompts (detailed in Appendix A.4) to improve the structure and interpretability of model outputs. We conducted inference on each task separately, and these model were seeded for each inference.

## 5 RESULTS

In this section, we present baseline results of selected MLLMs against the ground truth provided by the ASL native signers. Our research questions are: (1) how do the models leverage input from different modalities in our emotion recognition tasks, which we investigate through our conditions, (2) when there is a discrepancy, how do models fail, which we investigate by looking at the confusion matrices and models' reasoning, and finally (3) how do different model performances compare.

## 5.1 SENTIMENT ANALYSIS

Table 3 shows the sentiment analysis benchmark results. The video + caption condition yields the best performance for nearly all models on both the 3-class and 7-class sentiment tasks. When only

provided with the sign videos, models exhibited diverse biases and behaviors. AffectGPT consistently output sentiment as Neutral, suggesting an almost-complete lack of ability to recognize emotions in sign videos from visual cues alone. GPT-4o and Qwen2.5 tend to skew towards predicting positive sentiment, while we did not observe a consistent pattern for MiniGPT4. Caption-only performance was similar to or slightly better than video-only results with the exception of MiniGPT4, suggesting that models struggle to understand visual indicators of emotion in sign videos. However, the large performance gains in the video + caption condition across nearly all models and sentiment tasks demonstrates that visual information can contribute meaningfully to sign language sentiment analysis when integrated with textual context, highlighting the benefit of multimodal approaches.

The models, in general, exhibit a slight neutral-to-positive bias. Confusion matrices can be found in Appendix A.5.1. A possible reason for this is that many foundational models are pre-trained with an emphasis on being helpful, harmless and honest (Bai et al., 2022), leading to a neutral or positive bias to mitigate potentially harmful or incorrect assertions about a person's emotion state, especially in ambiguous contexts. However, more research is required to fully understand these observed model behaviors.

Table 3: Benchmark results of sentiment analysis. Bolded numbers: best score per column. All models achieved highest weighted Accuracy and F1 scores with video + caption inputs compared with uni-modality conditions, which suggest meaningful integration of information from both modalities.

| modality | model | sentiment (3-class) | | sentiment (7-class) | |
|---|---|---|---|---|---|
| | | wAcc | wF1 | wAcc | wF1 |
| Caption | MiniGPT4 | 1.92 | 5.92 | 0.00 | 0.00 |
| | Qwen2.5 | 37.74 | 33.72 | 18.13 | 12.94 |
| | AffectGPT | 36.80 | 44.91 | 15.67 | 9.99 |
| | GPT-4o | 31.12 | 49.53 | 15.13 | 18.23 |
| Video | MiniGPT4 | 34.68 | 40.00 | 14.46 | 13.03 |
| | Qwen2.5 | 27.34 | 16.47 | 10.26 | 2.44 |
| | AffectGPT | 33.33 | 0.04 | 14.29 | 0.04 |
| | GPT-4o | 40.72 | 24.43 | 19.81 | 5.97 |
| Video + Caption | MiniGPT4 | 21.65 | 36.89 | 9.76 | 12.18 |
| | Qwen2.5 | 41.10 | 54.29 | 15.84 | 14.51 |
| | AffectGPT | **56.18** | 64.37 | 21.02 | 16.13 |
| | GPT-4o | 52.13 | **76.72** | **22.89** | **26.35** |

## 5.2 SINGLE-LABEL EMOTION CLASSIFICATION

Table 4 shows the single-label emotion classification benchmark results. When only provided with the sign videos, models demonstrated limited ability to identify emotions beyond very broad and common categories. GPT-4o almost always classified videos as displaying either happiness or frustration, suggesting that it falls back to common emotional descriptors without the text as a contextual guide. Similarly, AffectGPT limited its predictions mostly to happiness, sadness, or neutral emotions, Qwen2.5 to happiness and neutral, and MiniGPT4 predominantly classified videos as happy.

Similar to the results of the sentiment analysis task, access to the video captions improved model performance, allowing for more accurate and nuanced emotional classification. Although GPT-4o still occasionally defaults to happiness and frustration, it shows enhanced capacity to distinguish emotions such as worry and disgust, and generally succeeded in identifying sentiment correctly. GPT-4o's tendency to favor labels such as worry and fear were aligned with the emotion co-occurrence patterns observed in the ground truth labels in EmoSign, suggesting that these emotions are relatively close together in the language embedding space (Huh et al., 2024). AffectGPT still retained its tendency to give neutral predictions, though less so than before. It occasionally confused frustration for happiness, but otherwise generally succeeded in identifying overall sentiment correctly. Qwen2.5 showed improved performance on the emotion classification task, but developed a tendency to predict frustration. Like AffectGPT, it occasionally confused frustration for happiness. MiniGPT4 continued to display a bias towards labeling videos as happy, even for videos with a ground truth of negative emotions such as disgust. Confusion matrices can be found in Appendix A.5.2. These persistent biases suggest the need for further model enhancements and fine-tuning on sign videos to improve their emotion recognition capabilities.

Unlike sentiment analysis, the caption-only condition showed similar and, sometimes, better performance to the video + caption condition in both wAcc and wF1 across models. We hypothesize that

Table 4: Benchmark results of single expression emotion classification. Bolded numbers: best score per column. Models performed similarly in caption-only and video+caption conditions and notably better than video as the only input, which suggest models rely on text information in this task.

| modality | model | HP Acc | SP(P) Acc | SP(N) Acc | WR Acc | SD Acc | FR Acc | DG Acc | FS Acc | AG Acc | NE Acc | total wAcc | total wF1 |
|---|---|---|---|---|---|---|---|---|---|---|---|---|---|
| Caption | MiniGPT4 | 59 | **50** | **67** | 0 | 60 | 0 | 14 | 13 | 33 | 11 | 27.01 | 34.16 |
| | Qwen2.5 | 70 | 20 | 0 | 36 | **70** | 14 | 40 | 0 | 0 | 55 | 30.49 | 43.09 |
| | AffectGPT | 76 | 20 | 17 | 43 | **70** | 29 | 0 | 11 | 33 | 18 | 31.37 | 43.09 |
| | GPT-4o | 87 | 40 | 14 | **86** | 40 | 29 | 30 | 53 | 33 | 0 | **41.16** | **55.89** |
| Video | MiniGPT4 | 69 | 20 | 14 | 0 | 0 | 0 | 0 | 0 | 0 | 27 | 13.01 | 22.02 |
| | Qwen2.5 | 35 | 0 | 43 | 0 | 0 | 0 | 0 | 5 | **33** | 27 | 14.39 | 18.53 |
| | AffectGPT | 11 | 0 | 0 | 7 | 30 | 0 | 0 | 5 | 0 | **73** | 12.62 | 11.03 |
| | GPT-4o | 35 | 0 | 0 | 7 | 0 | 0 | 20 | 53 | 0 | 0 | 11.50 | 20.76 |
| Video + Caption | MiniGPT4 | **89** | 0 | 14 | 7 | 30 | **43** | 10 | 0 | **33** | 9 | 23.56 | 35.89 |
| | Qwen2.5 | 63 | 40 | 29 | 64 | 0 | 14 | 20 | 32 | **33** | 55 | 34.96 | 44.67 |
| | AffectGPT | 85 | 20 | 0 | 50 | 30 | 14 | 10 | 32 | **33** | 27 | 30.17 | 47.77 |
| | GPT-4o | **89** | 20 | 0 | 79 | 20 | 29 | **50** | **74** | 0 | 0 | 35.97 | 55.09 |

HP: happiness; SP(P): surprise (positive); SP(N): surprise (negative); WR: worry; SD: sadness; FR: fear; DG: disgust; FS: frustration; AG: anger; NE: neutral.

the models may be overly reliant on textual captions, and unless the video modality clearly conveys the signers' emotion, it may introduce noise rather than improve predictions.

## 5.3 EMOTION CUE GROUNDING ANALYSIS

To obtain a preliminary understanding of model abilities to perform emotion cue grounding, we manually inspected several randomly selected videos alongside the ground truth and each model's corresponding reasoning outputs.

Without captions, MiniGPT4 and GPT-4o were still capable of identifying specific facial expressions within the sign videos and using them to reason about the signer's emotions, suggesting that these models can capture and interpret visual nuance to some extent. In contrast, AffectGPT and Qwen2.5 were only able to provide generic descriptions such as "The signer's facial expressions and body language do not reveal any obvious signs or strong emotion" (Figure 3).

With captions provided, the models appear to use the linguistic context to guide visual reasoning. As before, models were able to identify emotion cues such as hand gestures and posture. We verified that the cues recognized by GPT-4o, Qwen2.5 and MiniGPT4 (e.g., a thumbs-up sign) were truly present in the videos. However, there was a recurring sense that the models were attempting to construct explanations that were consistent with their judgment of the text sentiment, rather than independently recognizing emotions from visual cues, as the same cue could be interpreted in opposite ways between the Video-only and Video+Caption conditions (Figure 3). Despite general performance improvements with captions, we also observed several failure modes: models sometimes misinterpreted the text sentiment, correctly understood text sentiment but evaluated the visual inputs in ways that diverged from the Deaf annotators, or demonstrated poor understanding of sign language (e.g., claiming that they needed audio context to determine emotions).

GPT-4o frequently repeated statements such as "relaxed body language" and "generally positive sentiment", indicating a possible over-reliance on common language patterns without truly consulting the visual context. Qwen2.5 often highlighted the lack of audio and exhibited reluctance to make definitive statements about the signer's emotion. Its reasoning sometimes demonstrated a lack of understanding about sign language as a concept, with outputs such as "the exact content of the sign language cannot be determined without audio." As in the sentiment and emotion classification tasks, AffectGPT's reasoning displays a bias towards the neutral label, frequently repeating statements such as "neutral expression" and "lack of obvious body language cues." Like Qwen2.5, it was also hesitant to make statements about the signer's emotional state.

On the whole, we observed a significant performance gap between ground-truth labels and MLLM predictions, especially in the vision-only condition. This gap underscores the difficulty of the benchmark tasks as well as the current limitations of MLLMs in comprehending the nonverbal emotion cues in sign language. These findings align with prior work indicating that MLLMs often struggle with visual understanding, and that strong performance on visual question-answering tasks are

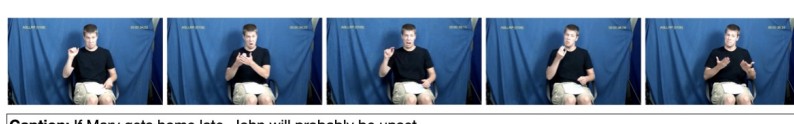

**Caption:** If Mary gets home late, John will probably be upset.
**Ground truth:** The mouth morpheme for 'oooh' conveys a sense of foreboding (A1). Emphasised 'arriving' [gets] and 'upset' with head shake and head tilt (A2).

**MiniGPT4:** John appears to be upset or frustrated due to Mary delaying returning home. His eyebrows are furrowed together, his lips curled downwards, as well as a slight clench of this jaw muscles.
**Qwen 2.5:** Based on the signer's hand gestures and body language, it appears that they are expressing a sense of concern or worry about the situation described in the video caption. The signer's facial expression seems to convey a sense of anticipation or anxiety.
**AffectGPT:** The visual cues from the facial expressions and body movements of the signer, along with the caption, suggest that the signer might be conveying a sense of worry or anxiety.
**GPT-4o:** The character's facial expressions and body language suggest concern and mild frustration, likely due to the potential for John to be upset. The signing speed and emphasis on certain words indicate a focus on the negative outcome of the situation.

**MiniGPT4:** The signer often exhibits a joyful expression during certain moments in the signature process. There were instances wherein excitement is palpable from their energetic gestures particularly when discussing thrilling topics like travel…
**Qwen 2.5:** Based on the signer's hand gestures and body language, it appears that they are expressing a sense of frustration or anger. The signer seems to be making emphatic movements with their hands, which could indicate a strong emotional response. However, without audio context, it's challenging to determine the exact nature of the emotion.
**AffectGPT:** The signer's facial expressions and body language do not reveal any obvious signs of strong emotion. The head is slightly tilted, possibly indicating interest or concentration in the signing action.
**GPT-4o:** The character maintains a neutral facial expression and steady signing speed throughout the clip. There are no significant changes in body language or facial expressions to indicate any strong emotions.

**Video + Caption**                                                **Video-only**

Figure 3: Sample emotion cue grounding outputs from each model. The highlighted sentences show how models can visually identify emotion cues, but interpret them differently based on whether the text caption is available for added context.

likely not due to genuine visual comprehension but rather a result of language shortcuts and spurious correlations with irrelevant visual information (Xiao et al., 2024).

## 6 LIMITATIONS AND FUTURE WORK

VADER offered a simple filter for emotionally salient videos; we found VADER results differed from the annotators' results often contained rich non-manual markers that conveyed emotions differently than the text, making them particularly valuable for training models to recognize visual emotional cues.

ASLLRP contains high quality recordings of signers from multiple POVs, but lacks more complex scenarios often present in the real world, such as multiple speakers. Future work can adapt our annotation pipeline to incorporate sign videos in more naturalistic settings. Further, future work can extract spatial and temporal information from the videos, such as optical flow and facial landmarks, which could provide critical context for the emotion cue grounding task Zhang et al. (2025).

In terms of benchmarking, we did not evaluate model performance on multi-label emotion classification tasks (Liu et al., 2022), which poses further challenge on current multimodal models' emotion and video understanding capabilities. In addition, as new open-sourced sign language recognition and translation models[1] become available, future work could investigate fine-tuning these models on our dataset to improve emotion reasoning within the context of sign languages and support more robust sign language recognition and translation.

## 7 CONCLUSION

In this paper, we introduced EmoSign, the first multimodal dataset containing sentiment and emotion labels specifically for ASL videos. By providing annotations from Deaf native ASL signers on sentiment, emotion categories, and detailed descriptions of emotion cues, EmoSign addresses a critical gap in both sign language research and emotion recognition. Our benchmark evaluations of several state-of-the-art multimodal LLMs revealed significant limitations in their ability to recognize emotions in sign language videos, particularly when relying solely on visual input without text captions. We hope that emotion-aware models would further improve the accuracy of sign language translation models, particularly for affective content, and encourage more research on the nuanced emotional expressions unique to sign languages.

---

[1] https://blog.google/technology/developers/google-ai-developer-updates-io-2025/

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

# A  APPENDIX

# A    APPENDIX / SUPPLEMENTAL MATERIAL

## A.1    DATA COLLECTION ADDITIONAL DETAILS

We employed a total of 3 annotators through a third-party vendor, specifically a full-service sign language interpreting and captioning company specializing in ASL. The annotators utilized the interface shown in Figure 4 for the annotation tasks. All three annotators completed all annotation tasks for all sign videos in the dataset. The vendor was paid $400 per annotator.

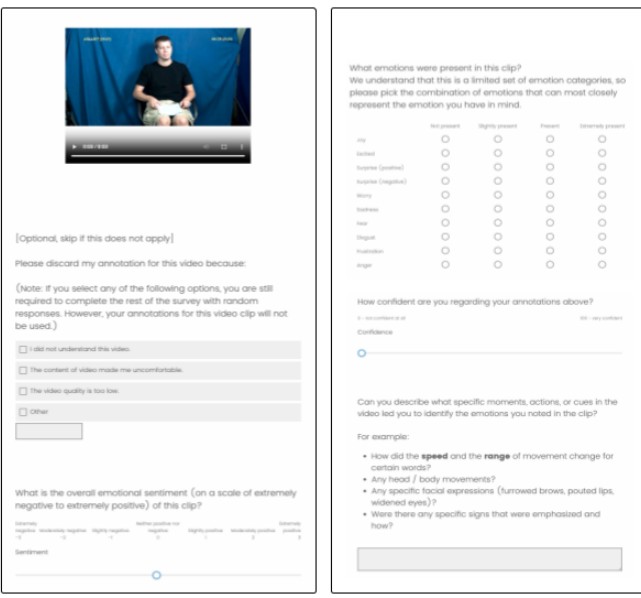

Figure 4: Annotation Interface

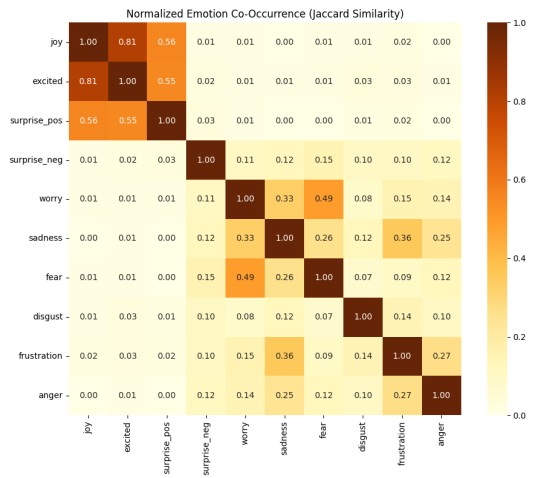

Figure 5: Jaccard Similarity of the original set of Emotion Labels.

## A.2 MODEL CARDS AND OUTPUTS

| Model | Link |
|-------|------|
| Qwen2.5-VL-7B-Instruct (Bai et al., 2025) | `https://huggingface.co/Qwen/Qwen2.5-VL-7B-Instruct/tree/main` |
| Qwen3-8B [2] | `https://huggingface.co/Qwen/Qwen3-8B` |
| AffectGPT (Lian et al., 2025) | `https://github.com/zeroQiaoba/AffectGPT` |
| MiniGPT4-video (Ataallah et al., 2024) | `https://github.com/Vision-CAIR/MiniGPT4-video` |

Table 5: Model cards for MLLMs used in EmoSign baseline results.

## A.3 GPT-4O DETAILS

The prompt template used for GPT-4o is as follows:

---

You are an expert in the field of emotions.

Please focus on facial expressions, body language, environmental cues, and events in the video and predict the emotional state of the character. Please ignore the character's identity. We uniformly sample 10 frames per second from this clip. Please consider the temporal relationship between these frames.

The video involves a person signing a sentence in ASL. You have three tasks:

Task 1 - On a scale of extremely negative (-3) to extremely positive (+3), what is the overall affective sentiment of this clip? Output your classification as a number between -3 and 3.

Task 2 - Which of the following affective categories were present in this clip? You may choose multiple options. The scale is from 0 to 3, where 0 is not present and 3 is extremely present. The possible categories are: joy, excited, surprise (positive), surprise (negative), worry, sadness, fear, disgust, frustration, anger. Output your classification as a json with the name of the emotion and a number between 0 and 3.

Task 3 - Describe what specific moments, actions, or cues in the video that led you to your responses above? (e.g., the speed at which the person signed, specific facial expressions, content of the sign, etc). Output your 2-sentence response as a string.

You should provide a structured response in the form of a Json string.

---

A sample response is formatted as follows:

---

"filename": "00ADU7t7IWI_1",
"Sentiment": "2",
"joy": "3",
"excited": "2",
...
"anger": "0",
"QA": "The person is smiling and signing with energetic movements, indicating a positive and engaging message. The consistent smile and lively hand gestures suggest joy and excitement."

---

## A.4 OTHER MLLM PROMPTS FOR SENTIMENT AND EMOTION RECOGNITION

The remaining MLLMs used for baseline evaluation struggled to follow the prompt template used for GPT-4o, which is relatively long and complex. As such, we created separate prompts for the sentiment analysis and single-label emotion recognition tasks presented in Section 5.. Several of the multimodal LLMs used in the analysis had a dedicated method for incorporating video captions, in which case we used it to pass the caption to the mode. Otherwise, we included the caption directly within the prompt.

---

**Sentiment Prompt:** You are an expert in the field of emotions. The video involves a person signing a sentence in ASL. Please focus only on facial expressions and body language of the signer in the video and try to recognize the emotional state of the signer. Your task is: on a 7 point Likert scale ranging from extremely negative to extremely positive, what is the overall affective sentiment of this clip? You must select an answer from this list: [Extremely Negative, Negative, Somewhat Negative, Neutral, Somewhat Positive, Positive, Extremely Positive].

**Single-label Emotion Recognition Prompt:** You are an expert in emotion analysis. Your task is to analyze the given video clip and select the SINGLE MOST DOMINANT emotion from the following list: [joy, excited, surprise (positive), surprise (negative), worry, sadness, fear, disgust, frustration, anger].

**Rules you MUST follow:**
1. **You must choose ONLY ONE emotion**—the one that is most prominent in the clip.
2. If multiple emotions seem present, select the strongest one.
3. If unsure, make your best judgment—do not leave the response blank.

**Penalties for not following instructions:**
- If you output more than one emotion, you will fail the task.
- If you refuse to answer or leave it blank, you will fail the task.

**Output format:** [selected emotion]

Now, analyze the clip and provide the SINGLE emotion label:

**Sentiment extraction prompt:** Based on this text, what is the sentiment of the signer? The text is: [inference output]. Keep your answer brief without any explanation.

**Emotion label extraction prompt:** Based on this text, what are the signer's emotions? The text is: [inference output]. Keep your answer brief without any explanation.

---

Figure 6 provides an example of a sign video and the associated MLLM outputs from our baseline experiments.

Figure 6: Sample frames from a sign video in the EmoSign dataset. The caption is: "If Mary gets home late, John will probably be upset."

**Qwen2.5 (caption + video):** Based on the signer's hand gestures and body language, it appears that they are expressing a sense of concern or worry about the situation described in the video caption. The signer's facial expression seems to convey a sense of anticipation or anxiety, which aligns with the idea of being upset if someone arrives late. Therefore, the most appropriate emotional state for the signer in this context would be "worry".

**Qwen2.5 (video):** Based on the signer's hand gestures and body language, it appears that they are expressing a sense of frustration or anger. The signer seems to be making emphatic movements with their hands, which could indicate a strong emotional response. However, without audio context, it's challenging to determine the exact nature of the emotion.

**Qwen2.5 (caption):** The caption suggests a potential negative outcome for John if Mary arrives home later than expected, which implies concern or worry about his feelings of disappointment or sadness.

**AffectGPT (caption + video):** The visual cues from the facial expressions and body movements of the signer, along with the caption "If Mary comes back late, John must be upset," suggest that the signer might be conveying a sense of worry or anxiety. This emotional state could be related to the potential impact of Mary's lateness on John. Therefore, the primary emotions displayed by the signer in the video might include worry, sadness, or anxiety.

**AffectGPT (video):** Given the video description, the signer's facial expressions and body language do not reveal any obvious signs of strong emotion. The head is slightly tilted, possibly indicating interest or concentration in the signing action. The eyes are closed, possibly to focus or concentrate on the signing, and the mouth is open, possibly due to speech or to express emotion. Considering these clues, it can be inferred that the signer might be in a calm or neutral emotional state. However, without additional information about the context of the signing or the individual's background, it is not possible to accurately determine the character's emotional state.

**AffectGPT (caption):** This sentence is a statement in the caption, not a question or inquiry. Based on the description of the neutral expression of the speaker in the video clue, as well as the lack of obvious emotional cues in the tone, it can be inferred that the speaker may feel indifferent or neutral towards this statement. Therefore, it is not possible to determine the speaker's emotional state, so a neutral emotion can be used as a general description.

**MiniGPT4-video (caption + video):** Mostly, John appears to be upset or frustrated due to Mary delaying returning home. His eyebrows are furrowed together, his lips curled downwards (shown by the wrinkles above them), as well as a slight clench of this jaw muscles which indicates stress or irritation - all these signs suggest he feels either anger/frustration (disgust) at her lateness. Secondarily one can see worry or anxiety expressed through brow wiggling while watching oneself being signing - there is an urge hidden behind those raised upper lids that could turn into full blown concern if things go wrong further...

**MiniGPT4-video (video):** The signer often exhibits a joyful expression during certain moments in the signature process. For example, when they finish signing an important sentence or document with satisfaction. They also show signs of happiness such as smiling wide and laughing occasionally which indicates positive emotions. There were instances wherein excitement is palpable from their energetic gestures particularly when discussing thrilling topics like travel experiences abroad or new technologies that arouse enthusiasm.

**MiniGPT4-video (caption):** The sentence implies that John will be upset if Mary gets home late which indicates a sense of concern for her safety and well-being. It suggests an expectation that he might experience negative emotions such as anxiety (worry) If something happens to someone they care about then it could also express sadness

**GPT-4o (caption + video):** The character's facial expressions and body language suggest concern and mild frustration, likely due to the potential for John to be upset. The signing speed and emphasis on certain words indicate a focus on the negative outcome of the situation.

**GPT-4o (video):** The character maintains a neutral facial expression and steady signing speed throughout the clip. There are no significant changes in body language or facial expressions to indicate any strong emotions.

**GPT-4o (caption):** The word 'upset' indicates a potential negative emotional response from John. The phrase 'probably be upset' suggests uncertainty and concern about the outcome, reflecting a sense of worry.

## A.5 ADDITIONAL DESCRIPTIVE STATISTICS OF THE DATASET AND BENCHMARK RESULTS

### A.5.1 ADDITIONAL SENTIMENT ANALYSIS RESULTS

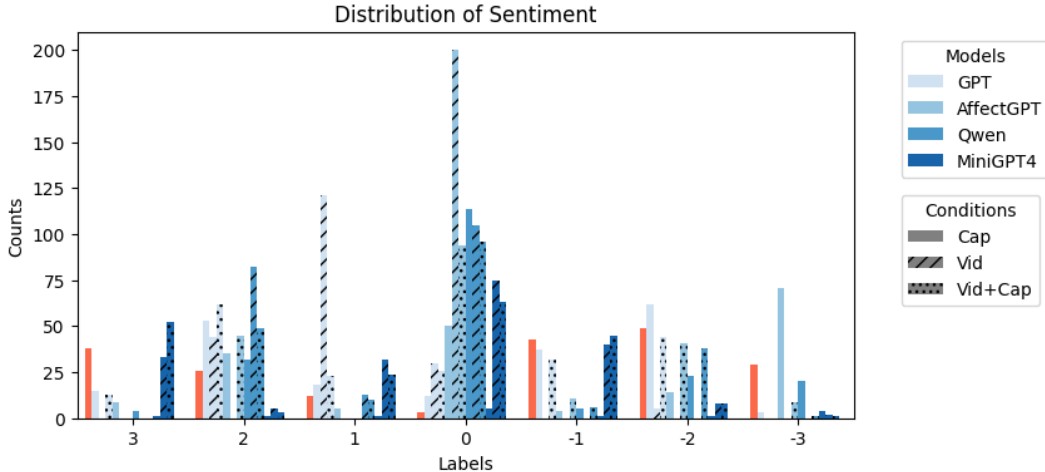

Figure 7: Distribution of Sentiment Analysis Results (7-class).

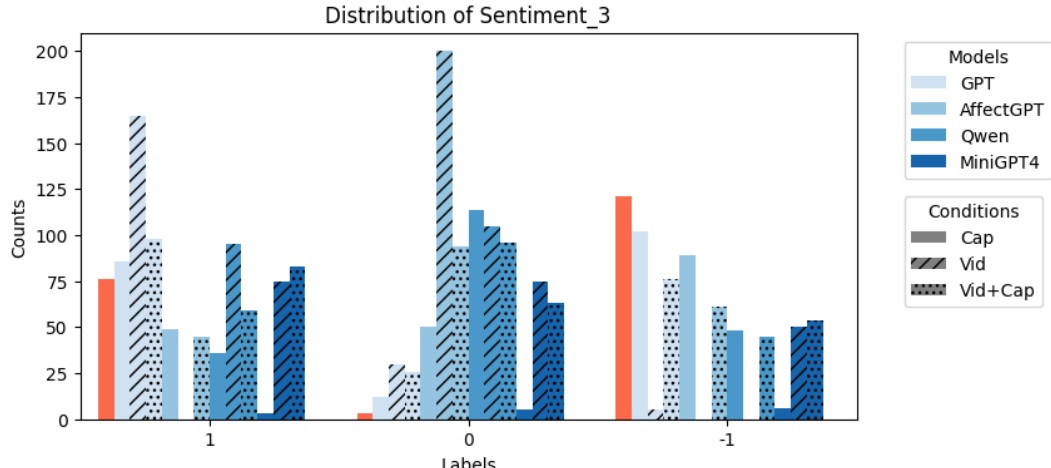

Figure 8: Distribution of Sentiment Analysis Results (3-class).

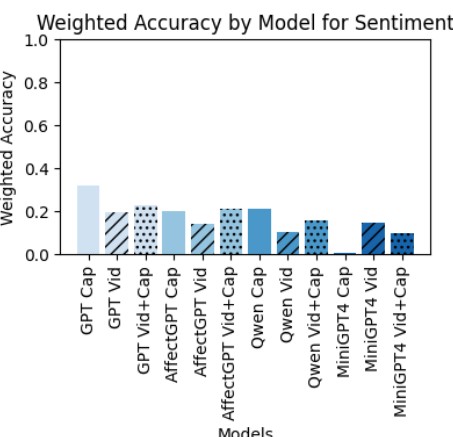

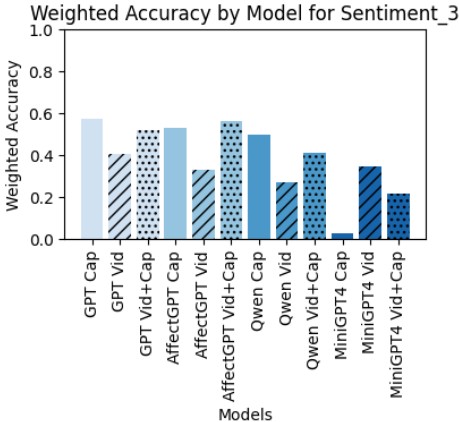

Figure 9: Weighted Accuracy of Models on Sentiment Analysis Task (7-class).

Figure 10: Weighted Accuracy of Models on Sentiment Analysis Task (3-class).

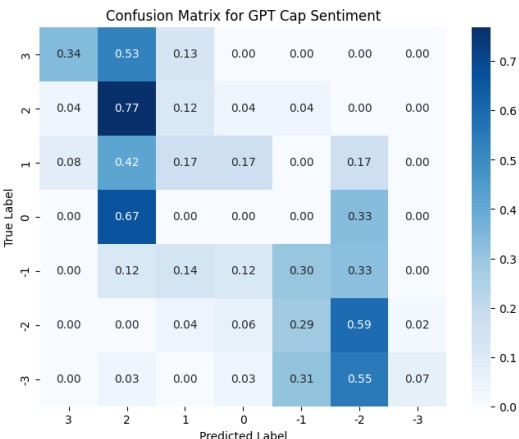

Figure 11: Confusion Matrix of GPT-4o (caption only) on Sentiment Analysis Task (7-class).

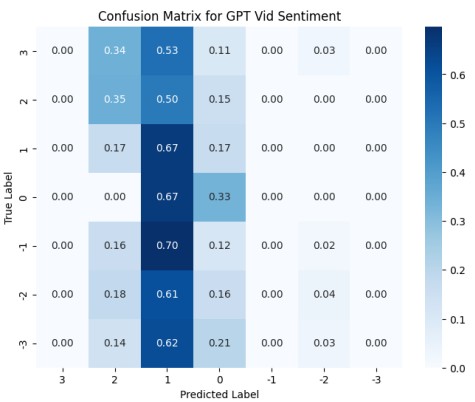

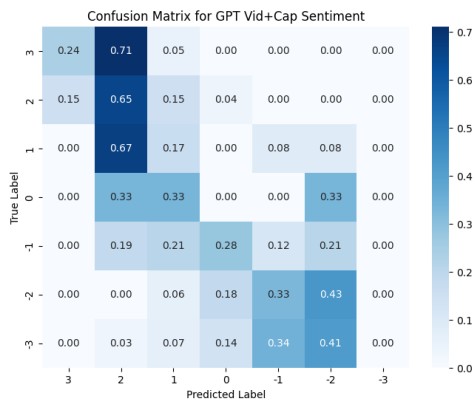

Figure 12: Confusion Matrix of GPT-4o (video only) output on Sentiment Analysis Task (7-class).

Figure 13: Confusion Matrix of GPT-4o (video + caption) output on Sentiment Analysis Task (7-class).

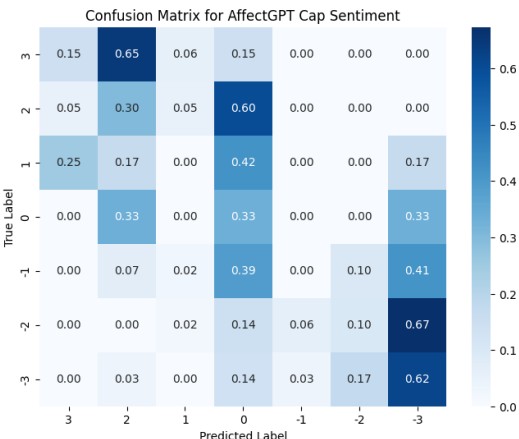

Figure 14: Confusion Matrix of AffectGPT (caption only) on Sentiment Analysis Task (7-class).

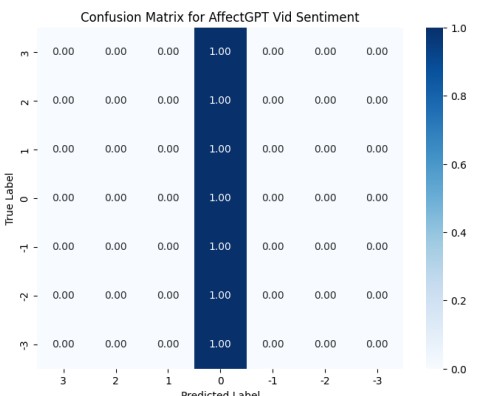

Figure 15: Confusion Matrix of AffectGPT (video only) output on Sentiment Analysis Task (7-class).

Figure 16: Confusion Matrix of AffectGPT (video + caption) output on Sentiment Analysis Task (7-class).

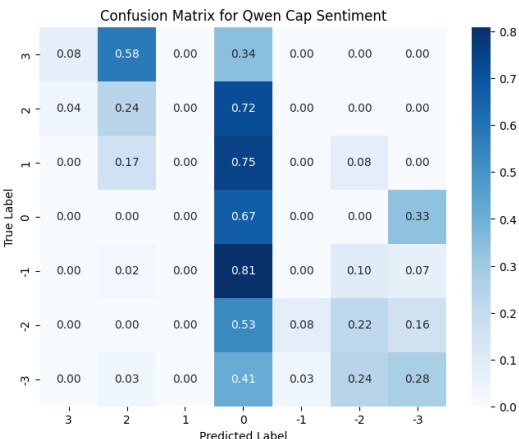

Figure 17: Confusion Matrix of Qwen-2.5 (caption only) on Sentiment Analysis Task (7-class).

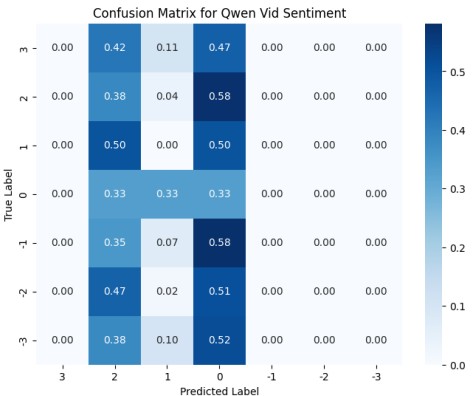

Figure 18: Confusion Matrix of Qwen-2.5 (video only) output on Sentiment Analysis Task (7-class).

Figure 19: Confusion Matrix of Qwen-2.5 (video + caption) output on Sentiment Analysis Task (7-class).

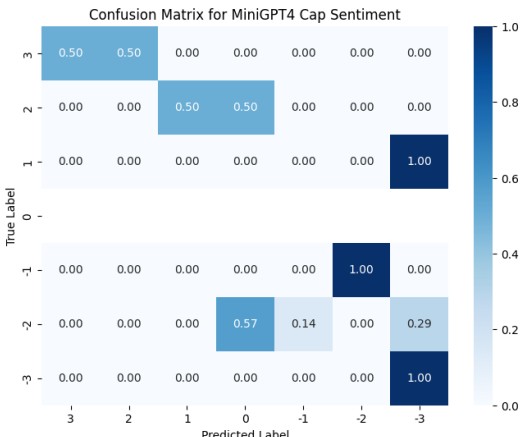

Figure 20: Confusion Matrix of MiniGPT4 (caption only) on Sentiment Analysis Task (7-class).

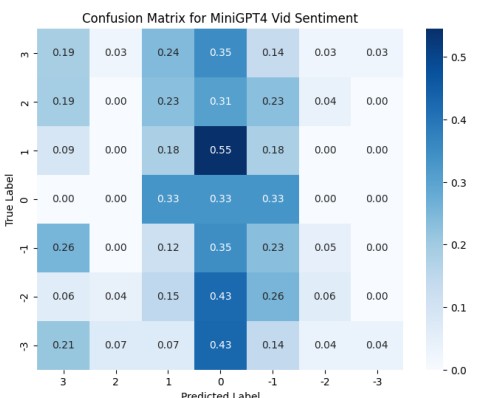

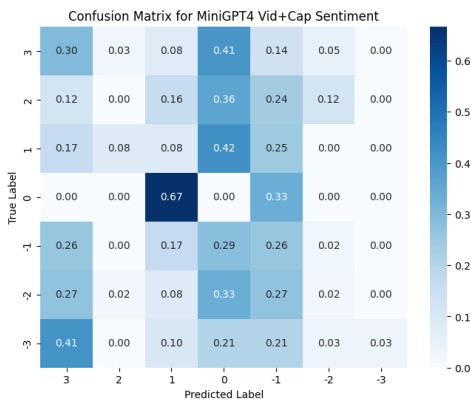

Figure 21: Confusion Matrix of MiniGPT4 (video only) output on Sentiment Analysis Task (7-class).

Figure 22: Confusion Matrix of MiniGPT4 (video + caption) output on Sentiment Analysis Task (7-class).

## A.5.2 ADDITIONAL EMOTION CLASSIFICATION RESULTS

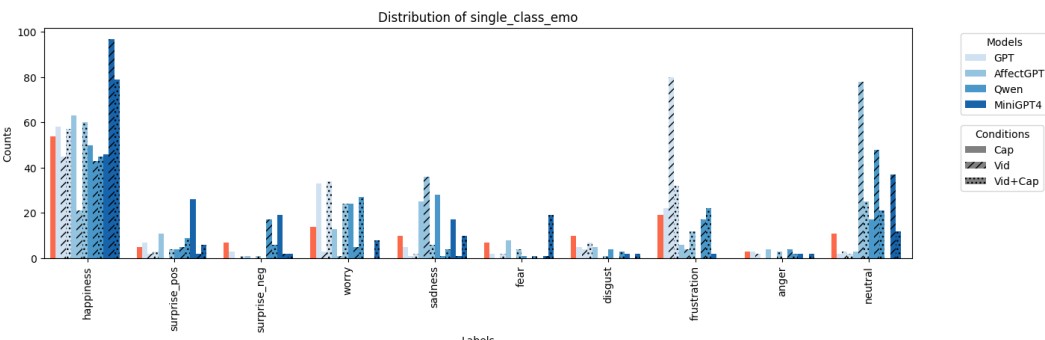

Figure 23: Distribution of Single Expression Emotion Classification Results.

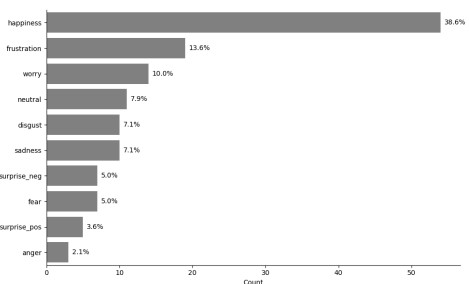

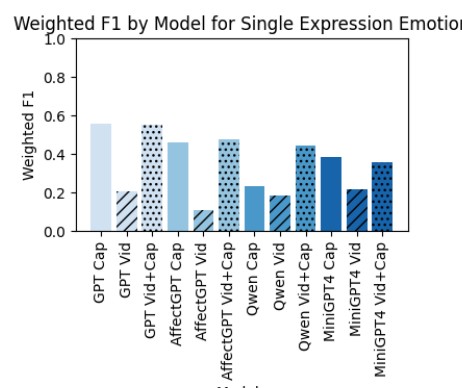

Figure 24: Distribution of emotion categories within the single expression set. Numbers above the bars indicate count.

Figure 25: Distribution of emotion categories within the multi expression set. Numbers above the bars indicate count.

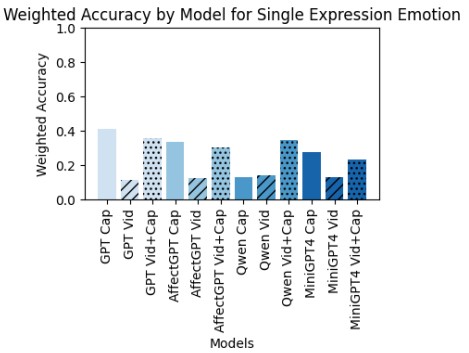

Figure 26: Weighted Accuracy of Models on Single Expression Emotion Classification Task.

Figure 27: Weighted Accuracy of Models on Single Expression Emotion Classification Task.

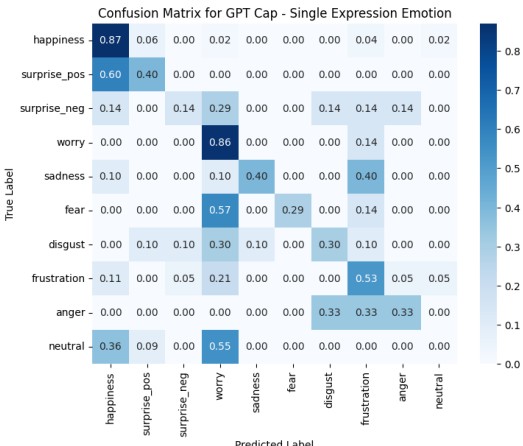

Figure 28: Confusion Matrix of GPT-4o (caption only) on Single Expression Emotion Classification Task.

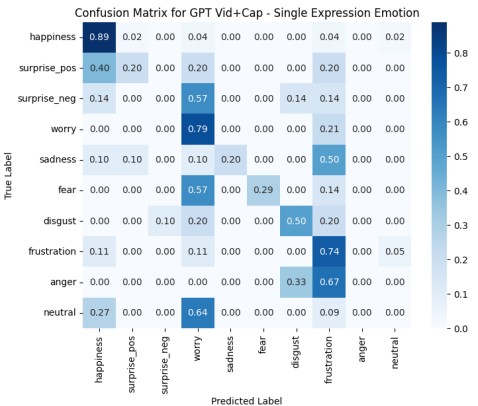

Figure 29: Confusion Matrix of GPT-4o (video only) output on Single Expression Emotion Classification Task.

Figure 30: Confusion Matrix of GPT-4o (video + caption) on Single Expression Emotion Classification Task.

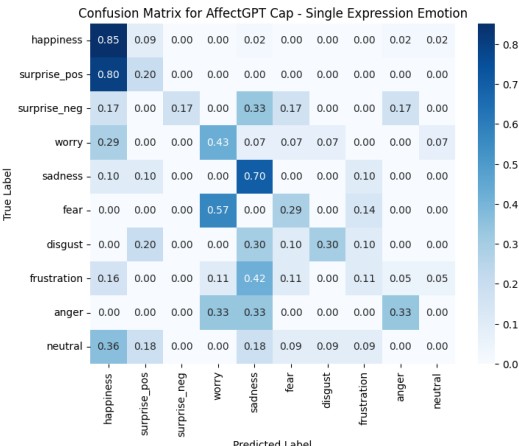

Figure 31: Confusion Matrix of AffectGPT (caption only) on Single Expression Emotion Classification Task.

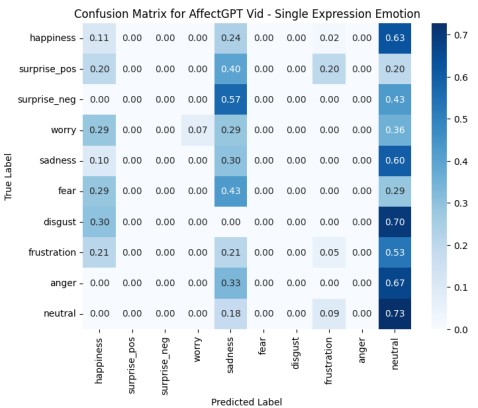

Figure 32: Confusion Matrix of AffectGPT (video only) output on Single Expression Emotion Classification Task.

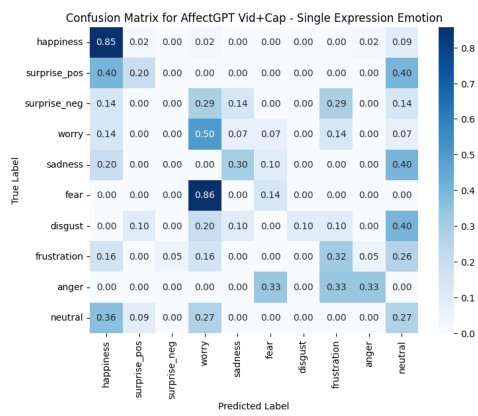

Figure 33: Confusion Matrix of AffectGPT (video + caption) output on Single Expression Emotion Classification Task.

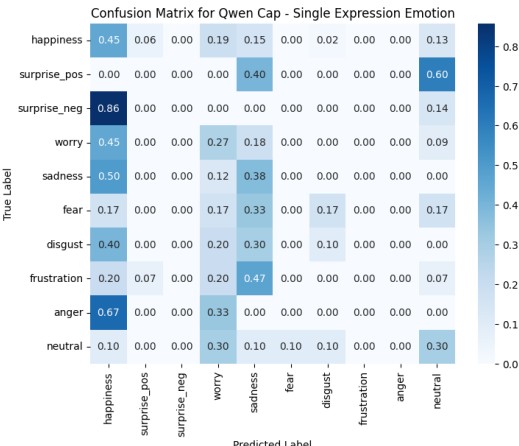

Figure 34: Confusion Matrix of Qwen-2.5 (caption only) on Single Expression Emotion Classification Task.

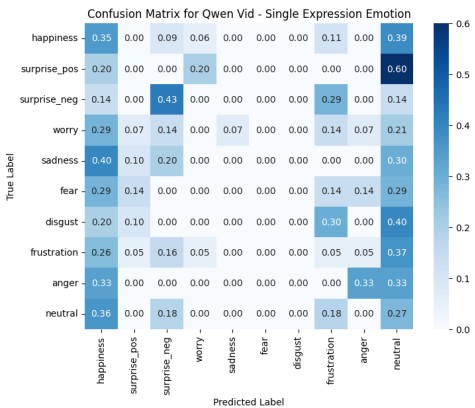

Figure 35: Confusion Matrix of Qwen-2.5 (video only) output on Single Expression Emotion Classification Task.

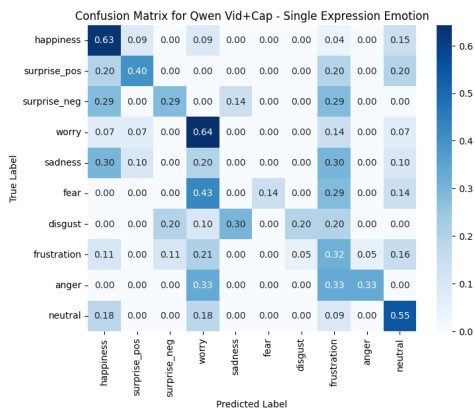

Figure 36: Confusion Matrix of Qwen-2.5 (video + caption) output on Single Expression Emotion Classification Task.

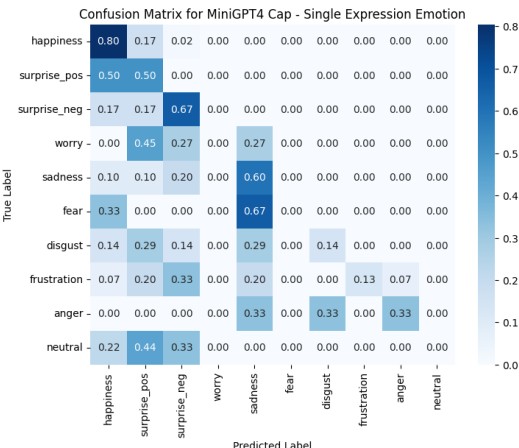

Figure 37: Confusion Matrix of MiniGPT4 (caption only) on Single Expression Emotion Classification Task.

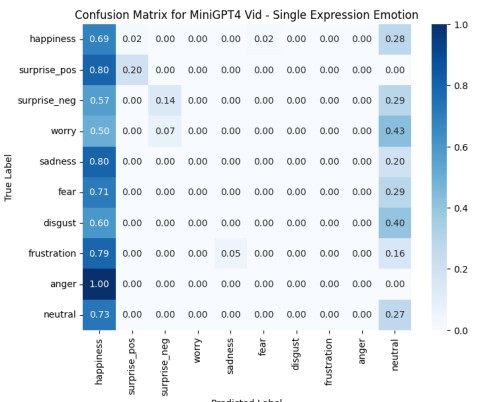

Figure 38: Confusion Matrix of MiniGPT4 (video only) output on Single Expression Emotion Classification Task.

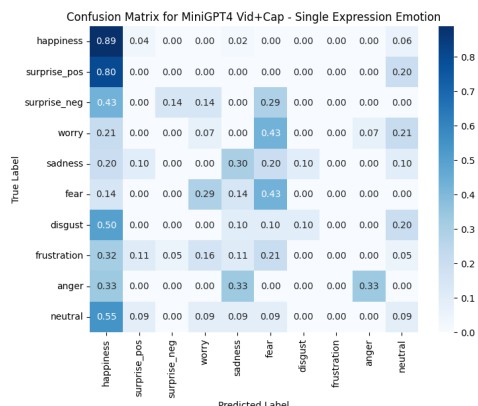

Figure 39: Confusion Matrix of MiniGPT4 (video + caption) output on Single Expression Emotion Classification Task.