# OpenReview forum: "EmoSign: A Multimodal Dataset for Understanding Emotions in American Sign Language"
_ICLR.cc/2026/Conference — Submitted to ICLR 2026_

### Official Review · Reviewer_LPZp · 2025-10-21

**Soundness:** 3
**Presentation:** 3
**Contribution:** 3
**Rating:** 6
**Confidence:** 3

**Summary:**

This paper is the first to present a dataset for emotion recognition in sign language videos, an as of yet underexplored area of sign language understanding/processing. This task is difficult for non-signers, suggesting it may also be difficult for neural networks. Three annotators with sign language knowledge annotate the data from an existing dataset, and the inter-annotater agreement is quite high. The authors then apply MLLMs without fine-tuning as baselines for this new dataset and find that they are unable to recognize emotions.

**Strengths:**

This paper introduces a new dataset for a task which was previously ignored in the field of sign language processing. It is a very useful work for a field which is starting to mature. The description of the dataset and how it was annotated is clear.

The paper is well written for the most part, and is easy to read and follow.

**Weaknesses:**

It would have been interesting to see the performance of a fine-tuned model, instead of only using off-the-shelf models. It makes sense that these models would not perform very well, because (1) they were never trained for similar tasks, (2) it's known they do not perform well for sign language tasks out of the box, except in certain cases such as SLT, and (3) non-signers have difficulty recognizing these emotions. Similarly, different models were evaluated with different prompts, making comparison difficult.

**Questions:**

Figure 1 has a typo: Consoliate - Consolidate.
Can you elaborate on annotator confidence? Is this per sample? Is it standard to have such a large range?
Why did you use a different temperature setting for GPT-4o compared to the other models? Why did you use different prompts?

---

### Official Review · Reviewer_KcKz · 2025-10-28

**Soundness:** 3
**Presentation:** 3
**Contribution:** 1
**Rating:** 2
**Confidence:** 5

**Summary:**

The paper EmoSign: A Multimodal Dataset for Understanding Emotions in American Sign Language introduces a new dataset aimed at addressing the gap in emotion recognition for American Sign Language (ASL). The dataset, EmoSign, includes 200 ASL video clips annotated with sentiment and emotion labels, with annotations provided by three Deaf ASL signers. The paper highlights the importance of facial expressions in ASL, asserting that they serve both grammatical and emotional functions, and presents the dataset as a valuable resource for training multimodal emotion recognition models. Through experiments, the paper demonstrates that existing models fail to integrate visual cues effectively and exhibit biases toward positive emotions. The authors suggest that this dataset can serve as a benchmark for future research in ASL emotion recognition.

**Strengths:**

1. Novelty in Data Collection: The EmoSign dataset is the first to focus specifically on emotion and sentiment analysis within ASL, providing an essential resource for the field.

2. Benchmarking Potential: The dataset presents a clear baseline for multimodal emotion recognition models, which is a useful resource for researchers looking to build on this area.

3. Multimodal Approach: The dataset takes into account both visual cues (e.g., facial expressions and hand gestures) and sentiment labels, offering a more comprehensive approach to emotion recognition than traditional text-only models.

**Weaknesses:**

1. Lack of Clear Novelty: While EmoSign is presented as a novel contribution, similar datasets (e.g., FePh) already exist in the domain of ASL emotion recognition. The paper does not clearly explain how EmoSign offers a significant advancement over these existing resources.

2. Inconsistency in Data Collection: The authors argue against using artificially recorded videos due to concerns over the "realism" of emotion expression, yet the dataset is based on pre-recorded videos from existing datasets, which presents a contradiction in their methodology.

3. Reliability of Expert Annotations: The dataset relies on annotations from three expert signers, which introduces a level of subjectivity. Emotional expression is inherently subjective, and variations in interpretation by different annotators may undermine the dataset’s generalizability and reliability.

4. Limited Experimental Insights: The paper mainly reports the failure of current multimodal models to handle emotion recognition in ASL videos, without offering actionable solutions or proposing novel model architectures to address these shortcomings. The experimental section lacks deeper exploration of model improvement or new methodologies.

5. Small Dataset Size: At only 16 minutes of total video length and 200 clips, the dataset may be too small to effectively train and evaluate deep learning models, which typically require larger datasets to perform well.

**Questions:**

1. Clarification of Novelty and Contribution

The paper claims that EmoSign is a novel contribution to emotion recognition in ASL. However, there are existing datasets, such as the FePh dataset, that also focus on emotional annotations in sign language. Could you clarify how EmoSign significantly advances the field compared to these existing datasets? What specific features or improvements make this dataset unique?

2. Data Collection Inconsistencies

In the paper, it is mentioned that you prefer natural, real-world recordings for emotion recognition tasks due to the perceived lack of authenticity in pre-recorded data. However, the EmoSign dataset uses pre-recorded ASL videos from other datasets. Could you explain this contradiction? Why did you choose pre-recorded data, and how does it align with your argument about the limitations of artificial recordings?

3. Reliability of Emotion Annotations

The dataset relies on annotations from three expert Deaf ASL signers. While these annotators bring valuable expertise, emotion annotations can be subjective and may vary between individuals. Could you provide more information on how you ensured the consistency and reliability of the annotations? Did you conduct any inter-annotator reliability tests (e.g., Krippendorff’s Alpha) to assess the consistency between annotators?

4. Model Performance and Further Exploration

The experimental results mainly focus on the failures of current multimodal models to recognize emotions in ASL videos. While these results are valuable, they do not offer insights into how to improve these models. What steps do you envision for improving model performance in future research? Do you plan to propose any novel architectures or methods for addressing the limitations you highlighted?

5. Dataset Size and Generalizability

Given that EmoSign consists of only 16 minutes of video and 200 clips, the dataset may be too small to effectively train and evaluate complex models. Could you elaborate on how you plan to scale the dataset in future work? Do you believe that a dataset of this size is sufficient for robust model training, or will you extend it in future iterations?

6. Handling Multimodal Data

The paper mentions the integration of both visual cues (facial expressions, hand gestures) and textual data (sentiment labels). Could you provide more details on how you plan to train models that can effectively combine these two modalities? What challenges do you foresee in developing multimodal models for ASL emotion recognition, and how do you plan to address them?

---

### Official Review · Reviewer_FK4v · 2025-10-30

**Soundness:** 2
**Presentation:** 3
**Contribution:** 2
**Rating:** 2
**Confidence:** 4

**Summary:**

This paper presents an ASL dataset called Emosign, which contains 200 utterances in overall 16 minutes of video playback.
This dataset is built from an existing dataset ASLLRP, and benchmarks three tasks: Emotion Classification, Emotion Cue Grounding, and Sentiment Analysis.
The main contribution of the dataset is the annotation, which costs three annotators (expert/native signer) each "roughly 3 hours".

**Strengths:**

Emotion expression in sign language translation is a long-standing topic. This paper establishes an emotion detection benchmark for ASL and evaluates the performance of four popular LMs on their benchmark.
The benchmark also comes with three different tasks and corresponding well-designed evaluation metrics with different input settings, including video, caption, and video+caption.
The paper is easy to understand.

**Weaknesses:**

1. The dataset is relatively small. And as all the data comes from only one public dataset, it may have domain issues. As such, the effectiveness of the benchmark is suspicious.
2. The paper discusses multimodal models in the abstract and introduction sections, but only evaluates several VLMs in the experiments. In SLT, many works have validated the effectiveness of facial input as well as body movement input. Should these works be included in the scope?
3. Continue from 1&2, only four tested models may not be enough to demonstrate the validity of the benchmark. Please consider more, like GPT-5.
4. The performance has already demonstrated that the models are not capable of giving reasonable results, especially in the video part, more like a random guess. So in this case, shouldn't the benchmark first make sure the models understand sign videos? Otherwise, it is difficult to diagnose the failure modes or justify the benchmark’s design goals.

**Questions:**

1. As stated in the paper, the Vader score is used as a primary selection method. What is the mapping between your human-labeled categories and the Vader score?
2. BTW, the current benchmark is very challenging. Would you consider having a simple version, e.g, only binary classification on positive and negative classes?
3. In your appendix, you mentioned Qwen3-8B in the table "Model cards for MLLMs used in EmoSign baseline results", but the related content is not shown in any part of the paper.

**Details Of Ethics Concerns:**

The benchmark uses an existing ASL video dataset called ASLLRP.
I have not seen any license/consent/agreements showing that the authors got permission to re-annotate the data and set it as a public benchmark. I only found the free-to-use statement for the annotating software in the ASLLRP paper.
Please help check, thanks!

---

### Official Review · Reviewer_dqi3 · 2025-10-31

**Soundness:** 3
**Presentation:** 3
**Contribution:** 2
**Rating:** 2
**Confidence:** 5

**Summary:**

EmoSign is a new set of emotion annotations on top of 16 minutes of videos from the ASLLRP dataset, targeting new emotion recognition/sentiment analysis tasks for ASL. There are a number of experiments and analysis performed on LLMs showing that they do not have a good grasp on emotions in sign language.

**Strengths:**

This is a relatively novel task that's especially interesting because facial expressions in sign languages are indeed often misunderstood (though I think typically in the opposite direction of the LLMs; I've seen people think signers look angry).

The paper is presented well and written in a way that meets best practices in the sign language field, including paying attention to data subject/annotator qualifications (Deaf native signers).

**Weaknesses:**

The two biggest problems with the paper to me are small dataset size and lack of motivation/insufficient baselines.

1.
16 minutes / 200 utterances is very small for an eval, when that artifact is essentially the contribution of the paper. (To be fair, if you're not hill-climbing against it, having some moderate amount of noise isn't terrible.) But the paper isn't framed as an "eval", it's a "dataset". And you don't see that it's 16 minutes long until Section 3 of the paper.

2.
The benchmark/task isn't motivated well / there are a lot of contradictory statements about its motivation. This is muddied by the fact that the only baselines are zero-shot LLMs (which have essentially no sign language capabilities, see e.g. https://arxiv.org/abs/2408.13585). They would be fine to include as a point of reference, but there really needs to be some kind of actual baseline trained on sign language data (and adapted to the task in some way, difficult when you only have 16 minutes of data!). It's possible that the task is trivial for models that have actually seen sign language; you can't tell from the paper. Or even if not trivial, I don't see a reason to think that it's orthogonal to translation benchmarks, and there's not much justification that emotion recognition is a practically useful task independent of its instrumental value to other tasks (vs. something like translation).

Some claims related to this that I found unjustified or contradictory:

> "Existing American Sign Language (ASL) datasets 1 focus primarily on translation tasks with English captions, lacking the emotional annotations necessary for
developing and evaluating affect recognition systems."

There's no reason to think emotion annotations are "necessary" to develop such systems except perhaps to specify a particular output format, maybe to evaluate them (but what are sign language affect recognition systems useful for?).


> "The poor performance observed in vision-only
condition suggests a need for specialized visual encoders trained specifically on non-verbal emotional expression, rather than relying on general-purpose vision models. Additionally, our ablation
experiments motivate the development of architectures that prevent text from overwhelming visual
information"

This seems like the general-purpose vision models just need to be trained better (more diverse kinds of data, and especially video data!). I see no reason that they need specialization per se or different architectures.


This implies that EmoSign is necessary because translation benchmarks aren't sufficient:
> "Unlike existing sign language
datasets that focus primarily on translation capabilities, EmoSign specifically targets the affective
dimensions of signing. Our benchmark results with 4 multimodal models show that current models
fail to integrate visual cues and heavily rely on text captions for emotion reasoning."

But then this said that better emotion understanding improves translation quality (aka would be measured by translation benchmarks, so why do you need this different benchmark?):
> "promising results for spoken languages, where incorporating emotional understanding improves performance on affective content
recognition and translation tasks"


A smaller comment:

It looks like there is at least one (recent) prior work on sentiment analysis in sign language senhttps://ieeexplore.ieee.org/document/10601084 , maybe more I'm not aware of, though it is certainly not well-studied.

**Questions:**

I could maybe have my mind changed on the motivation for the benchmark, but probably only up to a 4. I do think you really need proper sign language model baselines, which is a pretty substantial thing to add in a rebuttal.

More of a nit, and you do talk about this a little bit on lines 91-93, but I think there is more you could say about whether this is really "emotion recognition", which usually implies you're trying to infer the state of mind of someone as opposed to some expression/demeanor that they're putting on for linguistic/communicative purposes. "Sentiment analysis" completely makes sense.

---

### Meta-Review · Area_Chair_hnGJ · 2025-12-03

**Summary:**

The authors introduce a new multi-modal benchmark for sign language recognition. The data set contains visual cues like hand gestures as well as sentiment labels to allow for emotion recognition in the context of sign language recognition and thus brings together two interesting modalities. The authors also report on empirical results of four multi-modal LLMs on their data.

**Reviewer Concerns:**

The reviewers are thrilled by the combination of the modalities and find this approach in general very useful. However, they consider the size of the new benchmark with its 16 minutes containing 200 utterances (extracted from the same source) too small for a state-of-the-art benchmark. They also criticise the four evaluated models as insufficient in number and diversity of the tested approaches (only VLMs; there is no dedicated sign-language model (e.g., evaluated on only a single modality for comparison), etc.).

**Reviewer Scores:**

There is no rebuttal that could have possibly changed the reviewers opinion on the paper.

---

### Decision · Program_Chairs · 2026-01-26

Reject